# Shearography-Based Near-Surface Defect Detection in Composite Materials: A Spatiotemporal Object Detection Neural Network Trained Only with Simulated Data

**DOI:** 10.3390/nano15070523

**Published:** 2025-03-30

**Authors:** Guanlin Li, Yao Hu, Hao Wang, Qun Hao, Yu Zhang

**Affiliations:** 1School of Optics and Photonics, Beijing Institute of Technology, Beijing 100081, China; 3120230613@bit.edu.cn (G.L.);; 2National Key Laboratory on Near-Surface Detection, Beijing 100012, China; 3Systems Engineering Research Institute, China State Shipbuilding Corporation Limited, Beijing 100094, China

**Keywords:** shearography, near-surface defect detection, spatiotemporal object detection network

## Abstract

Shearography is a non-destructive defect detection technique that, when combined with neural networks, can efficiently and accurately detect near-surface defects in composite materials. However, the high cost of the dataset significantly limits the application of neural networks in shearography. Current simulation data generation techniques fail to eliminate the discrepancies between simulated and experimental data, resulting in suboptimal performance when training neural networks with only simulated data. To address this issue, this paper utilizes phase map sequences measured by shearography as the medium for defect detection and designs a YOWO_SS3D spatiotemporal object detection network. The network simultaneously learns both the spatial distribution features and temporal variation patterns of simulated phase map sequences, achieving high-accuracy detection of defects. The experimental results show that, with only 4000 frames of simulated data for training, our network achieved a detection accuracy of 96.99% on experimental phase maps, which is considerably higher than the 65.37% accuracy achieved by training the YOLOv4 network with the same simulated data. Using our technique, only pre-generated simulation data are required to train the network, enabling YOWO_SS3D to be directly deployed for practical defect detection tasks. This approach eliminates the substantial costs associated with collecting experimental training data and promotes the application of neural network technology in the shearography field.

## 1. Introduction

In recent years, composite materials have been widely used in aerospace, automotive, wind energy, and marine industries due to their superior mechanical properties. However, near-surface defects such as delaminations, debonding, and cracks often occur during manufacturing and service, compromising structural integrity [1]. Among numerous near-surface defect detection methods, shearography has emerged as an outstanding optical inspection technique for composite materials, owing to its advantages of non-destructive testing and strong anti-interference capabilities [2,3,4].

Shearography works by applying an external force to the specimen, causing out-of-plane displacement. The wrapped phase distribution of the gradient of this out-of-plane displacement is the final measurement result of the shearography, called the phase map [5]. Near-surface defects appear in the phase map as abnormal phase fringes in a butterfly shape. Therefore, locating these abnormal fringes in the phase map enables defect detection. However, manually detecting defects from phase maps is time-consuming and labor-intensive, which has prompted many researchers to focus on automatically detecting defects from phase maps.

Earlier, scholars commonly used morphological image processing techniques for the automatic detection of near-surface defects in shearography. For instance, in 2014, Wang [6] applied derivatives to phase maps and used morphological methods to detect defects from the derivative images. In 2017, Revel [7] employed wavelet transform techniques to locate defects in phase maps. These morphological detection methods often involve setting multiple empirical parameters based on the test phase map, which can affect their robustness and accuracy. 

As a result, more and more researchers have turned to neural networks to assist shearography in near-surface defect detection. In 2019, Chang [2] used the fast R-CNN network to detect bubble defects in tires, training the model with 325 experimental images collected from experiments, and the final accuracy was approximately 89%. In 2022, Nagmy [8], with the assistance of an automotive company, obtained 793 bubble-free tire inspection images and 207 images with bubbles. They used a texture feature extraction network for binary classification of tire bubble detection results. In the same year, Chang [3], with the help of a local tire manufacturer, acquired 3450 phase maps of tire bubbles. They used YOLOv3 combined with incremental learning, achieving an accuracy of 92.2%. The aforementioned teams all use single-frame phase maps as input to the network and train and test the network using real experimental data. Since phase map acquisition is difficult, the datasets used in the above studies are generally small, and only with the help of large companies, thousands of labeled phase maps can be obtained to support neural network training. The limited amount of data directly restricts the defect detection performance of each team. The difficulty of acquiring large quantities of experimental shearography datasets is a common problem, which severely limits the application of neural network technology in near-surface defect detection using shearography.

To address the issue of creating shearography datasets, many researchers have used simulation techniques to generate data and then mixed the simulated data with experimental data for network training. In 2022, Li [9] used finite element simulation technology to simulate shearography, generating 5117 sets of simulated phase maps. They mixed the simulated data with 250 experimental images and trained a YOLOv4 model, achieving an accuracy of 85%. The team leveraged simulation technology to augment the experimental data, overcoming the overfitting problem in small-sample scenarios. However, due to the differences between simulated and experimental data, they still required hundreds of experimental images for mixed training. The accuracy achieved using only simulated data for training was merely 43%. In 2024, Wang Yonghong [10] leveraged the similarity between electronic speckle pattern interferometry (ESPI) [11] and shearography to simulate phase maps on the phase maps obtained from ESPI experiments. This resulted in 4950 sets of quasi-real data, and by using a semantic segmentation network, they achieved an accuracy of 99.15% on the validation set. This approach of simulating phase maps from ESPI data is both innovative and effective. However, this dataset generation method still relies on actual ESPI experiments and is essentially an experimental data augmentation algorithm. Overall, although many methods exist to simulate phase maps, current approaches still only consider the spatial features of a single-frame phase map. A major challenge for neural networks is the significant gap between simulated and experimental data. This discrepancy arises because simulation data do not account for the inherent characteristics of specific hardware, such as the difficulty of accurately modeling complex material properties and environmental noise. As a result, networks trained solely on simulation data often exhibit lower accuracy on experimental data; for example, the network may struggle to distinguish between noise and actual defects in experimental data, leading to missed or false detections. This limitation necessitates a heavy reliance on experimental data for training. However, collecting experimental data that comprehensively covers various test specimen materials, shapes, and defect types is challenging. This difficulty has become a major constraint on the generalization ability of deep neural networks.

To eliminate the reliance on experimental data during the neural network training, while ensuring detection accuracy and reducing the cost of creating shearography datasets, this paper proposes a defect detection method using a spatiotemporal object detection network. This method learns defect features from both the spatial and temporal dimensions, effectively eliminating the interference caused by random noise, allowing high-performance defect detection with simulation data alone for training. Although the similarity between single-frame simulated and experimental phase maps is low, their temporal features in the phase map sequences, which represent continuous experimental results under dynamic external forces in shearography experiments, are very similar. Therefore, leveraging the temporal features of phase map sequences for defect detection is the core idea of our approach. First, we designed the YOWO_SS3D (You Only Watch Once: Shearography Skip 3D) neural network to extract features from input phase map sequences. YOWO_SS3D consists of a 3D backbone network and a 2D backbone network, which extract the temporal features from the phase map sequence and the spatial features from each phase map frame. We also designed the detection fusion head with a Skip3D feature fusion module to integrate these two types of features. Second, we constructed a simulation dataset of phase maps in the form of video sequences and trained the YOWO_SS3D network using it, with different video sampling intervals and ranges. Then, we built an experimental dataset for the test. The highest accuracy achieved by YOWO_SS3D on the experimental dataset was 96.99%, demonstrating the powerful defect detection capability of this method. Ablation experiments were conducted on the 3D backbone network module and the Skip3D feature fusion module, validating the significant performance improvement brought by both modules. Finally, we conducted comparative experiments using the classic YOLOv4 network for low-data training and simulation–experiment data hybrid training, proving the advantages of the YOWO_SS3D network over traditional deep learning methods.

The core contributions of this paper are summarized as follows.

Proposing a high-accuracy defect detection method utilizing temporal features to further enhance the spatial features of the defects. This approach captures the temporal similarities between simulated and experimental phase-map sequences, enabling the neural network to learn experimental characteristics of the defects from simulations. As a result, even when trained solely on simulation data, the proposed method achieves higher detection accuracy than the traditional method, which focuses only on the spatial feature of the defects in a phase map.Designing the YOWO_SS3D defect detection network consists of a 3D backbone network and a 2D backbone network, which extract the temporal features from the phase map sequence and the spatial features from each phase map frame. Additionally, we designed the Skip3D feature fusion module to integrate these two types of features. Compared to YOLOv4 [9] and other object detection networks that only contain a 2D backbone to extract spatial features, YOWO_SS3D leverages its 2D + 3D backbone architecture to extract spatial features and temporal features, allowing it to effectively distinguish between noise and defects. With this advanced network architecture, YOWO_SS3D demonstrates exceptional defect detection performance. Furthermore, even when trained solely on simulation data, YOWO_SS3D achieves high-accuracy defect detection on experimental data.Providing a simulation approach for generating phase map sequences, which is achieved by mathematically modeling the entire shearography process. The modeling results accurately reflect the phase maps of defect regions observed in real experiments, making them suitable for network training.

## 2. Materials and Methods

In Section 2.1, we introduce the basic principles of shearography and explain the approach of using simulated phase map sequences for training. From Section 2.2 to Section 2.4, we introduce the model, dataset, and training strategy of the network, respectively.

### 2.1. Shearography Spatiotemporal Object Detection Method

Shearography measures the out-of-plane displacement gradient of the tested specimen by applying an external force, with its phase distribution referred to as a phase map. Assuming that at time t, light with a wavelength λ is incident on a region experiencing an out-of-plane displacement w_t_, it undergoes diffuse reflection and forms lateral shearing interference within the optical system. The resulting phase map φ_t_ is determined by the phase distribution caused by the out-of-plane displacement and the phase shift introduced by lateral shearing. Thus, the phase difference is proportional to both the partial derivative of the out-of-plane displacement $\partial w_{t}/\partial d$ and the shearing amount |d|, with 4π/λ as the conversion factor between displacement and the reflected phase, as shown in Equation (1) [5]:(1)$$\varphi_{t}=\frac{4\pi }{\lambda }\cdot \frac{\partial w_{t}}{\partial d}\cdot \left|d\right|$$

Under the influence of external forces, the defected areas of the tested specimen generate unique out-of-plane deformation gradients, which are reflected in the phase map as butterfly-shaped anomalous fringes. Therefore, performing anomalous fringe detection on the phase map enables defect detection.

Figure 1a shows the phase map obtained from an actual shearography experiment, which contains three defects. Due to factors such as environmental noise, uneven external force distribution, optical alignment errors, and errors in spatial carrier phase extraction [12], the phase map obtained from the actual experiment often exhibits phenomena such as blurred or broken fringes, as shown in the red regions of Figure 1a. Using material mechanics formulas similar to those in reference [9], the phase map can be simulated, as shown in Figure 1b. The simulated phase map represents an ideal state, where the anomalous fringes in the defect areas are very clear, with no noise or fringe blurring and breakage. While single-frame simulation data can, to some extent, approximate the general outline of the anomalous fringes in the experimental data, it is difficult to realistically reproduce the fringe blurring and breakage, so defect detection using only simulated data to train a standard object detection network results in poor performance.

According to Equation (1), the phase map is related to the deformation of the tested specimen, and the deformation is influenced by the magnitude of the external force. Therefore, if the external force applied to the tested specimen in the shearography experiment is gradually increased continuously, the anomalous fringes in the phase map will become denser, exhibiting dynamic changes such as fringe densification. By gradually increasing the external force within the time interval [0,*T*] and dynamically detecting the tested specimen, a phase map sequence $\varphi \left(t\right)$ can be obtained, as shown in Equation (2):(2)$$\varphi \left(t\right)=\left\{\varphi_{t}\left|0\leq t\leq T\right.\right\}$$

Figure 1c shows a sequence of actual phase maps obtained from measurements, while Figure 1d displays the phase map sequence obtained using the material mechanics formula simulation method. Although each frame of the simulated data still differs from the experimental data, from the perspective of the entire time axis, both sequences exhibit phase fringes in the defect areas that change dynamically with variations in the applied external force. This temporal feature is consistent across both datasets.

Therefore, leveraging the temporal features can, to some extent, improve the similarity between the simulated and experimental data. Furthermore, in a phase map sequence, only the magnitude of the external force changes, and the defect distribution across all video frames remains the same. Thus, for simulated phase map sequences, although each frame’s spatial features differ from the experimental data, they still serve as the primary source of information. By appropriately combining the spatial and temporal features, the performance of the model trained with simulated data can be improved.

### 2.2. Model

Based on the method in Section 2.1, combining the spatial and temporal features of phase map sequences to jointly train the network model can improve the defect detection performance. Based on this idea, we constructed a YOWO_SS3D network, based on the traditional spatiotemporal object detection network framework of YOWO [13,14], specifically for shearography. This network follows a spatial feature-dominant, temporal feature-assisted approach. The overall structure of the network is shown in Figure 2, and it mainly consists of a 2D backbone module, a 3D backbone module, and a detection fusion head.

In the 2D backbone network module, the FreeYOLO_2D [14] backbone network is used to extract spatial features from each single-frame phase map, resulting in classification features and regression features, denoted as $F_{2D\_\mathrm{cls}}$ and $F_{2D\_\mathrm{reg}}$, respectively. The 2D backbone network effectively performs feature extraction from the defect areas in each phase map frame, capturing spatial features such as the phase fringe contours and abrupt changes in fringe intensity. These spatial features help accurately distinguish defects from the background and noise.

In the 3D backbone network module, the shufflenetv2_3D [15] backbone network is used to extract spatiotemporal features from the phase map sequence, resulting in spatiotemporal features $F_{3D\_\mathrm{ST}}$. The 3D backbone network can focus on the dynamic changes in the phase fringes of defect areas across adjacent frames, effectively capturing the motion of fringe densification, which is a commonality between simulated and experimental data. This enables the network to effectively learn the features of experimental data through simulated data.

Finally, we designed the detection fusion head module to combine the spatial and temporal features, yielding the final defect detection results. The structure of the detection fusion head module is shown in Figure 3. The classification features $F_{2D\_\mathrm{cls}}$ extracted by the 2D backbone network are first concatenated with the spatiotemporal features $F_{3D\_\mathrm{ST}}$ from the 3D backbone network, resulting in a combined feature. After two convolution operations, the first step of feature fusion is completed. Afterward, the classification feature $F_{2D\_\mathrm{cls}}$ is concatenated with a zero matrix, and similarly, after two convolution operations, it is directly added to $F_{\mathrm{cls}\left|\mathrm{ST}\right.}$, completing the fusion of the classification features. This shortcut path that bypasses the 3D temporal features is referred to as Skip3D. This residual-like connection provides a direct shortcut for the spatial features, allowing them to bypass the temporal features. It ensures the dominance of spatial features in the phase map sequence, preventing redundant temporal features from causing negative optimization. After completing the feature fusion, we introduce an attention mechanism. The feature map obtained from the channel fusion module is reshaped into two-dimensional feature maps. These two feature maps are element-wise multiplied, and the attention matrix is obtained through the SoftMax function. The attention matrix is then element-wise multiplied with the feature map, resulting in the output feature map from the channel encoder. The same process is applied to the regression features extracted by the 2D backbone network. Finally, the output results include the defect category, bounding box, and confidence, which are denoted as Cls, Reg, and Conf, respectively.

### 2.3. Dataset

To test the spatiotemporal object detection performance using only simulated data for training, we constructed separate simulation and experimental datasets. The simulated data are used as the training and validation set for the neural network, while the experimental data are used to evaluate the final defect detection accuracy of the network and conduct comparison experiments. For each frame of the phase map, the smallest enclosing rectangle around the defect is used as the defect coordinate label, and the defect category is uniformly labeled as “defect”.

#### 2.3.1. Simulation Dataset

According to Equation (1), the phase map is related to the derivative of the out-of-plane deformation of the tested specimen. Therefore, the deformation of the tested specimen under external force can be numerically modeled from the perspective of material mechanics, allowing for the simulation of phase maps and facilitating the rapid and large-scale generation of shearography datasets. The commonly used external force loading method in shearography is pressure loading [16], where the specimen is fixed around the edges and subjected to pressure as shown in Figure 4a. For the defect area, the deformation above the near-surface defect can be approximated as a plate under pure bending and twisting with fixed edges [17], and the deformation of the defect area can be analyzed using plate theory [18], as shown in Figure 4b. For the entire tested specimen, if the impact of near-surface defects is neglected, the deformation of the whole specimen can also be approximated as a plate under pure bending and twisting with fixed edges. Similarly, the overall deformation of the tested specimen can be analyzed using plate theory, as shown in Figure 4c.

According to elasticity theory and plate theory, the differential equation for the deformation at the defect location is given by Equation (3) [18]:(3)$$\left\{\begin{array}{l} \nabla^{2}\nabla^{2}w=\frac{F}{D}\\ \nabla^{2}=\frac{\partial^{2}w}{\partial x^{2}}+\frac{\partial^{2}w}{\partial y^{2}}\\ D=\frac{Eh^{3}}{12(1-\nu^{2})} \end{array}\right.$$
where w is the surface displacement, F is the external force, D is the bending stiffness, $\nabla^{2}$ is the Laplace operator in 2D space, E is the Young’s modulus, h is the depth of the defect below the surface, and ν is the Poisson’s ratio. The solution to the system of Equation (3) depends on the boundary conditions [18] of the defect. Therefore, we classify the defect shapes into circular and rectangular forms; the parameters are defined as shown in Figure 5.

The tested specimen is subjected to loading through air pressure, with the radius of the circular defect denoted as *r* and the dimensions of the rectangular defect as *a* and *b*. The approximate solutions for the surface displacement $w_{\mathrm{cir}}$ and $w_{\mathrm{rec}}$ for the circular and rectangular defects, calculated from the boundary conditions, are given by Equations (4) and (5) as follows:(4)$$w_{\mathrm{cir}}=\frac{F}{64D}{\left[r^{2}-(x^{2}+y^{2})\right]}^{2}$$(5)$$\left\{\begin{array}{l} w(x,y)={\mathrm{Acos}}^{2}(\frac{\pi x}{2a}\;)\cos^{2}(\frac{\pi y}{2b}\;)\\ A=\frac{16Fa^{4}b^{4}}{\pi^{4}D[3(a^{4}+b^{4})+2a^{2}b^{2}]} \end{array}\right.$$

According to Equation (1), the phase map is related to the derivative of the deformation along the shearing direction. Therefore, by taking the partial derivative of the surface displacement along the $\left(d_{x},d_{y}\right)$ shearing direction and substituting it into Equation (1), the relationship between the abnormal fringes and the external force can be derived, as shown in Equations (6) and (7):(6)$$\varphi_{\mathrm{cir}}=\frac{-\pi \left|d\right|F}{4\lambda D}\cdot \left(r^{2}-\left(x^{2}+y^{2}\right)\right)\cdot \left(xd_{x}+yd_{y}\right)$$(7)$$\left\{\begin{array}{l} \varphi_{\mathrm{rec}}=-\frac{4A\pi^{2}\left|d\right|}{\lambda }\cdot \left(\frac{\cos\left(\frac{\pi x}{2a}\right)\sin\left(\frac{\pi x}{2a}\right)\cos^{2}\left(\frac{\pi y}{2b}\right)}{a}d_{x}+\frac{\cos^{2}\left(\frac{\pi x}{2a}\right)\cos\left(\frac{\pi y}{2b}\right)\sin\left(\frac{\pi y}{2b}\right)}{b}d_{y}\right)\\ a=\frac{16Fa^{4}b^{4}}{\pi^{4}d[3(a^{4}+b^{4})+2a^{2}b^{2}]} \end{array}\right.$$
where $\varphi_{\mathrm{cir}}$ and $\varphi_{\mathrm{rec}}$ represent the phase maps of the circular defect and the rectangular defect, respectively. Taking the material mechanical parameters of epoxy resin as an example, with an external force of 3 kPa and a shearing amount of 3 mm, and wrapping the calculated phase within the range of $\left[-\pi ,\;\pi \right]$, the simulated abnormal fringe patterns for the circular defect $\varphi_{\mathrm{cir}}$ and the rectangular defect $\varphi_{\mathrm{rec}}$ can be obtained, as shown in Figure 6a,b.

Based on Equations (6) and (7), using the material mechanical parameters of epoxy resin as an example, we randomly adjust the size and position of the defects. Each frame of the simulated phase map has a resolution of 224 × 224, with two circular defects and two rectangular defects in each frame, and “defect” is used as the dataset label. By adjusting the temporal sampling interval and sampling range of the simulated phase map sequence, different simulation datasets can be generated for network training.

#### 2.3.2. Experimental Dataset

In order to collect experimental phase map and analyze the performance of the network trained on simulated data, we set up the Michelson-type shearography system for dynamic detection as shown in Figure 7a. In this setup, the laser wavelength is 532 nm, and a slit aperture with a width of 0.8 mm is selected to improve the signal-to-noise ratio when extracting the spatial carrier phase. The imaging lens used is LANO-FA7528M23-2M with a focal length of 75 mm, and the CMOS camera is BFS-U3-200S6C-C with a pixel size of 2.4 µm. The shearing amount was set to 6 mm. The test specimen, which includes square and circular defects representing common issues such as voids and delaminations, was subjected to uniform air pressure loading using a pneumatic pump. To protect the specimen from plastic deformation, the external force sampling range was limited to 0–100 mmHg. To increase the diversity of the experimental data, the shear phase map sampling intervals were randomly distributed between 0.8 and 2.6 mmHg, with an average sampling interval of 1.4 mmHg. A total of 3216 frames of phase maps with a resolution of 1024 × 1024 were obtained, as shown in Figure 7b,c. After manually labeling the defects, 1500 frames were used for mixed training comparison experiments, while 1716 frames were reserved as the test set to evaluate the model’s defect detection performance.

### 2.4. Training Strategy

We used the classic YOWO loss function [14], as shown in Equation (8):(8)$$\begin{array}{l} L\left(\left\{a_{x,y}\right\},\left\{b_{x,y}\right\},\left\{c_{x,y}\right\}\right)\\ =\frac{1}{\mathrm{Npos}}\sum_{x,y}L_{\mathrm{conf}}\left({\overset{\wedge }{c}}_{x,y},c_{x,y}\right)\\ +\frac{1}{\mathrm{Npos}}\sum_{x,y}I_{\left\{{\overset{\wedge }{a}}_{x,y}>0\right\}}L_{\mathrm{cls}}\left({\overset{\wedge }{a}}_{x,y},a_{x,y}\right)\\ +\frac{5}{\mathrm{Npos}}\sum_{x,y}I_{\left\{{\overset{\wedge }{a}}_{x,y}>0\right\}}L_{\mathrm{reg}}\left({\overset{\wedge }{b}}_{x,y},b_{x,y}\right) \end{array}$$
where $L_{\mathrm{conf}}$ and $L_{\mathrm{cls}}$ represent binary cross-entropy loss functions, $L_{\mathrm{reg}}$ is the GIou loss [19], and $a_{x,y}$,$b_{x,y}$,$c_{x,y}$ correspond to classification prediction, regression prediction, and confidence prediction, respectively. ${\overset{\wedge }{a}}_{x,y}$,${\overset{\wedge }{b}}_{x,y}$,${\overset{\wedge }{c}}_{x,y}$ represent the baseline experimental values, and Npos denotes the number of positive samples. The training parameters are shown in Table 1. The batch size for each training epoch was set to 8, and the Adam optimizer along with the classic YOWO system loss function was used. In the first training epoch, a linear warm-up of the learning rate was applied, with a base learning rate of 1 × 10^−4^. The linear warm-up strategy [20] gradually increases the learning rate in the early stages of training, which helps improve the model’s stability and convergence speed, while also avoiding risks such as gradient explosion and overfitting, thereby enhancing the overall training effectiveness. Subsequently, in epochs 2–6, a multi-step learning rate decay was applied, reducing the learning rate by half after each epoch. The multi-step decay dynamically adjusts the learning rate during key stages of the training process, helping to accelerate convergence, prevent overfitting, and improve the model’s stability and generalization ability. Finally, in the 7th epoch, the learning rate was fixed at 6 × 10^−6^ for fine-tuning the model.

The model’s performance on the test set is represented by accuracy (ACC), and its calculation formula is shown in Equation (9):(9)$$\mathrm{ACC}=\frac{TP}{TP+FP+FN}\times 100\%$$
where TP represents true positives, FP represents false positives, and FN represents false negatives. The ACC is calculated by averaging the results of each frame’s prediction. A higher ACC value indicates better performance in detecting abnormal fringes in the phase maps.

## 3. Results and Discussion

To evaluate the performance of the proposed YOWO_SS3D network, we trained the network using simulation data and first performed accuracy validation on the simulation test datasets, as described in Section 3.1. Then, the detection accuracy was validated with the experimental test datasets, and the impact of the sampling interval and sampling range of the training simulation data was discussed, as described in Section 3.2. Two sets of ablation experiments were designed to verify the significant improvement in network performance brought about by the 3D backbone network and the Skip3D feature fusion module, as presented in Section 3.3. Additionally, we designed two comparison experiments to demonstrate the advantages of YOWO_SS3D over traditional methods, as shown in Section 3.4. All training experiments were conducted on a GPU device, specifically the RTX 4090D with 24 GB of VRAM.

### 3.1. Simulation Data Testing and Results

The YOWO_SS3D network was trained and evaluated using the simulation technique described in Section 2.3.1. We set the sampling interval of the phase map sequence to 5 mmHg, with a sampling range of 0–100 mmHg, resulting in 300 sequences (6000 frames) of phase map. Among them, 200 sequences (4000 frames) of simulated phase map were used for the training dataset, 50 sequences (1000 frames) for the validation dataset, and 50 sequences (1000 frames) for the test dataset. The loss function curve after training is shown in Figure 8a. At the end of the loss function curve, both the validation and training loss curves follow the same downward trend, and there is no significant difference between the loss values of the validation and training sets, indicating that the network training did not overfit. The defect detection results of the trained YOWO_SS3D network on the simulated data test set are shown in Figure 8b. Since defect detection on the simulation data is relatively simple, the YOWO_SS3D network achieved an ACC of 100% on the test dataset.

### 3.2. Experimental Data Testing and Results

To verify the defect detection performance of the YOWO_SS3D model on experimental data, which was trained solely on simulation data, we input 11 sequences (1716 frames) of experimental phase map obtained in Section 2.3.2 into the trained YOWO_SS3D network. The network achieved an average ACC of 96.99% on the 1716 frames. Figure 9 illustrates a typical set of defect detection results with gradually increasing external force applied to the test specimen. In Figure 9a, the external force is relatively small, and the deformation of some of the test specimens is not significant. The abnormal fringes are sparse, and the abnormal fringes of the small defect on the right are almost undetectable. However, the YOWO_SS3D network successfully detected the defect. To make the small defects more visible, we applied a larger external force to the test specimen. This also caused the phase fringes of the large defects to become very dense and blurry, even completely losing the black-and-white phase fringe features, as shown in the left and upper defects in Figure 9c. At this point, the difference between the experimental data and the simulation data is substantial. Despite this, the YOWO_SS3D network, trained solely on simulation data, still accurately detected the defects.

We tested the impact of the sampling interval and sampling range of the simulation data on the network performance. Seven sub-datasets with different sampling intervals and five sub-datasets with different sampling ranges were designed. Each sub-dataset contained 5000 frames of simulation data, and the data were split into training and validation sets in an 8:2 ratio. After training the YOWO_SS3D network, accuracy tests were performed on the experimental dataset. The experimental results are shown in Table 2.

According to Table 2, the sampling interval and sampling range of the simulation data have a slight impact on the defect detection performance of the YOWO_SS3D network. Nevertheless, the accuracy remains 95.65% ± 1.33%, indicating that the YOWO_SS3D network, trained solely on simulation data, not only ensures high performance but also possesses the advantage of being less sensitive to the video parameters of the training dataset, which facilitates the creation of simulation datasets for industrial applications.

In our training tests, YOWO_SS3D performed best on the simulation dataset with a sampling frequency of 5 mmHg and a sampling range of 0–100 mmHg. Therefore, this simulation dataset was selected for network training in subsequent experimental tests.

### 3.3. Ablation Experiments

To verify the effectiveness of the 3D backbone network module and the Skip3D feature fusion module when training the YOWO_SS3D network on simulation data, we designed the following two sets of ablation experiments.

#### 3.3.1. Ablation Experiment for the 3D Backbone Network Module

To verify the contribution of the 3D backbone network module in the YOWO_SS3D network, we set the 3D weights in the feature fusion stage to 0, while keeping the other network structures unchanged, resulting in the YOWO_SS3D(2D) model. This modification effectively decouples spatiotemporal interactions, forcing the model to rely solely on spatial features. We trained both the YOWO_SS3D(2D) and YOWO_SS3D networks on the same simulation dataset following the training process outlined in Table 1. After training, we tested both models on a 1000-frame simulation test dataset and a 1716-frame experimental test dataset. The results are shown in Table 3.

According to Table 3, comparing Groups A and B, after removing the 3D backbone, the network’s performance on the simulation dataset remains close to 100% accuracy. This indicates that the YOWO_SS3D(2D) network successfully learned the spatial information of single-frame simulation data. However, the accuracy of YOWO_SS3D(2D) on the experimental data is only 53.48%, which is similar to the accuracy of conventional single-frame object detection networks trained on simulation data, as reported in reference [8]. This is significantly lower than the 96.99% accuracy achieved by YOWO_SS3D. This ablation experiment demonstrates that the temporal features extracted by the 3D backbone network play a crucial role in significantly improving the performance of a network trained solely on simulation data.

Partial visualization results of defect detection for experiment Groups A and B are shown in Figure 10. The first row shows the ground truth of the defects, while rows a and b represent the defect detection results for the two groups, A and B, as shown in Table 3. The first column shows the simulation data results, and the second, third, and fourth columns correspond to experimental data results. The red boxes highlight the defect detection results obtained by the YOWO_SS3D model presented in this study. 

By comparing the results, it is clear that, although both networks achieve nearly 100% accuracy on the simulation test dataset after training with the same simulation data, a significant difference exists in their detection results on the experimental test dataset. The YOWO_SS3D network effectively avoids false positives and missed detections by leveraging the temporal feature of fringe densification. In contrast, the YOWO_SS3D(2D) network, which only utilizes spatial features, has difficulty distinguishing between defects and background noise that resembles broken fringes, as well as identifying defects with broken fringes. This results in a significant number of false positives in the YOWO_SS3D(2D) network, such as the detection results at the edges of the images in Figure 10b2–b4. The ablation experiment clearly demonstrates that when trained on simulated data, the YOWO_SS3D network with the 3D module maintains high accuracy even on experimental data, despite the significant differences between simulated and experimental data. This is because the distinction between randomly varying noise and systematically changing defects is highly pronounced in temporal features. As a result, YOWO_SS3D, which leverages temporal features, achieves robust detection performance. This experiment proves the substantial benefit of temporal features for networks trained solely on simulated data. 

#### 3.3.2. Ablation Experiment for the Skip3D Feature Fusion Module

To further analyze the impact of the Skip3D feature fusion module in the detection fusion head of the YOWO_SS3D network, we conducted an ablation experiment on the Skip3D module. In this experiment, we removed the Skip3D module from the YOWO_SS3D network, causing the features extracted by the 2D backbone network and the 3D backbone network to be directly fused. This resulted in the YOWO_SS3D(Skip3D_test) network. Following the training parameters in Table 1, we trained the YOWO_SS3D(Skip3D_test) network on a 4000-frame simulation data training set and tested its accuracy on both the simulation test dataset and the experimental test dataset. The results are shown in Table 4.

By comparing the results of Groups A and B, the YOWO_SS3D network with the Skip3D module demonstrates stronger defect detection performance on the experimental data. The Skip3D module improved the defect detection accuracy on the experimental data by 2.22%. This indicates that the design of the Skip3D feature fusion module is well-suited to the characteristics of shearography. The spatial feature-dominant, temporal feature-assisted spatiotemporal object detection approach brought by the Skip3D module is indeed effective, enhancing the defect detection performance of the network trained with shearography simulation data.

### 3.4. Comparative Experiment

To evaluate the performance of the YOWO_SS3D network in both limited data training and hybrid training scenarios compared to traditional methods, we reproduced the method [9] using a hybrid-trained YOLOv4 with simulation-experimental data for shearography defect detection. Additionally, we removed the temporal labels from the entire video frames of the phase map sequence and treated them as a standard object detection dataset for training the YOLOv4 network. We designed experiments with limited data training and simulation-experimental hybrid training. Under the condition that the amount of data used for training remained the same, we compared the performance differences between YOWO_SS3D and YOLOv4.

#### 3.4.1. Comparison Experiment with Limited Data Training

In practical shearography defect detection tasks, the amount of experimental data is often limited. To analyze the defect detection performance of the YOWO_SS3D network with small amounts of experimental data, we compared it with the classic YOLOv4 network. We used only experimental data for training, with Experiment Groups A to E gradually increasing the data volume. The training and validation sets were split at a ratio of 8:2. Both YOLOv4 and YOWO_SS3D networks were thoroughly trained, and their performance was evaluated using a 1715-frame experimental test dataset. The results are shown in Table 5.

In Table 5, Group A uses only 100 frames of data for network training. Under such a small data condition, the accuracy of YOWO_SS3D is significantly higher than that of YOLOv4. As the data volume increases, the accuracy of both networks shows an upward trend, but YOWO_SS3D consistently outperforms YOLOv4. In the range where 100 to 1500 frames of data are used for training, the maximum accuracy of YOLOv4 is 87.31%, while the maximum accuracy of YOWO_SS3D is 89.53%. We believe that the superior performance of YOWO_SS3D in the experiments can be attributed to its advanced network architecture and unique learning mechanism. In contrast, YOLOv4 is a typical 2D network that lacks the ability to learn temporal features. When trained solely on simulation data, YOLOv4 is more susceptible to noise interference, leading to a high rate of missed and false detections. By effectively leveraging both spatial and temporal features, even with limited data, YOWO_SS3D demonstrates significantly superior defect detection capabilities compared to YOLOv4.

#### 3.4.2. Comparison Experiment of Simulation–Experimental Data Hybrid Training

To compare the performance of YOWO_SS3D and the traditional YOLOv4 network during simulation–experimental data hybrid training, we designed the following comparison experiment. Referring to the dataset division strategy from reference [9], we trained the networks using 4000 frames of simulation data and gradually added experimental data starting from 0 for hybrid training. Finally, we tested the performance of both YOLOv4 and YOWO_SS3D networks on a 1715-frame experimental data test set. The results are shown in Table 6.

In Table 6, Experiment Group A uses only simulation data for training, resulting in relatively poor performance for the YOLOv4 network, with an accuracy of only 65.37%. As experimental data are added to the training, the performance of the YOLOv4 network gradually improves, significantly surpassing the performance observed in Table 4 when only experimental data are used. This demonstrates that hybrid training with simulation–experimental data can effectively improve the performance of object detection networks under limited data conditions. This result is consistent with the conclusions in reference [9]. On the other hand, the YOWO_SS3D network achieved an accuracy of 96.99% when trained with only simulation data. During simulation–experimental data hybrid training, its detection accuracy remained stable around 95%, consistently outperforming YOLOv4. This indicates that YOWO_SS3D has superior defect detection performance compared to traditional single-frame defect detection networks.

### 3.5. Discussion

Based on experimental analysis, the YOWO_SS3D network demonstrates excellent performance in multiple aspects after being trained on simulated data. We believe that the YOWO_SS3D network holds significant potential for practical engineering applications, provided that the following aspects are emphasized. First, the optical devices used in practical engineering should have the capability for multi-frame continuous measurement to provide phase map sequences for detection. For example, using the spatial carrier method [21] for dynamic phase map calculation is recommended. Additionally, phase shifting method [22] can be employed if the deformation of the test object is controllable and stable, and the process is performed as quickly as possible. Second, the types of defects that may occur in the actual test objects should be estimated in advance to ensure that corresponding defect categories are included in the simulated dataset. Finally, to ensure optimal detection performance, the video parameters of the simulated dataset should be carefully aligned with those of the actual measurement process, minimizing any discrepancies that might arise. While this work demonstrates the potential of YOWO_SS3D, it does not include evaluations on complex defects (such as crack defects) or robustness testing in real industrial scenarios with multi-material substrates and surface interference. These aspects can be explored in future research to strengthen the practical applicability of the proposed method.

## 4. Conclusions

To address the issue of limited availability of shearography datasets, this paper designs the YOWO_SS3D spatiotemporal object detection network, which combines the temporal features of phase map sequences with spatial features to perform near-surface defect detection. This approach achieves high-performance deep learning defect detection using only simulation data for training. A Skip3D feature fusion module is designed specifically for the characteristics of phase map sequences, enabling the effective fusion of temporal and spatial features. Additionally, the material mechanics simulation formula for shearography is derived to realize the simulation of phase map sequences. We conducted extensive experiments using the test specimen, which includes square and circular defects representing common issues such as voids and delaminations. The results show that after training YOWO_SS3D with 4000 frames of simulated phase map sequences, the model achieved an accuracy of 96.99% on the experimental test dataset, significantly outperforming YOLOv4 trained on the same simulated dataset (65.37%) and surpassing YOLOv4 trained on the hybrid simulated–experimental dataset (90.74%). Ablation experiments focusing on the 3D backbone and the Skip3D feature fusion module confirm their contributions to the accuracy improvement of the YOWO_SS3D network when trained on simulation data. The YOWO_SS3D network proposed in this paper can perform near-surface defect detection on experimental data by solely relying on simulation data for training, significantly reducing the difficulty of generating shearography datasets and lowering the entry barrier for the application of neural network technology in this field. To extend the practical impact of this work, two critical directions should be prioritized in the future: (1) The test specimen model in this study is a simplified version, with defect types limited to only circular and square shapes. Future work should evaluate the model’s performance in handling defects with complex stress distributions (e.g., crack-type defects). (2) Validating detection robustness in real industrial environments involving multi-material substrates and surface interference.

## Figures and Tables

**Figure 1 nanomaterials-15-00523-f001:**
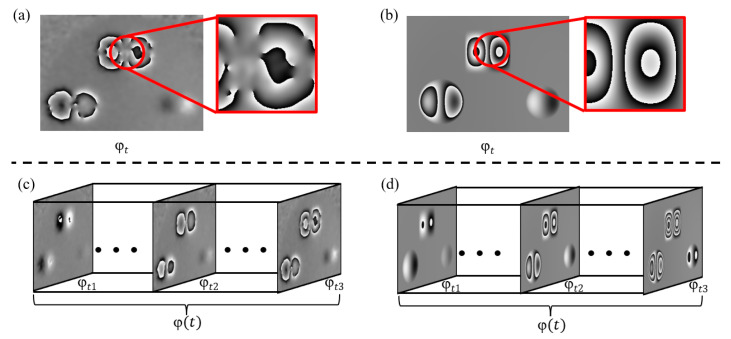
Examples of phase maps. (**a**) Phase map at time t from experimental data. (**b**) Phase map at time t from simulated data. (**c**) Phase map sequence from experimental data. (**d**) Phase map sequence from simulated data.

**Figure 2 nanomaterials-15-00523-f002:**
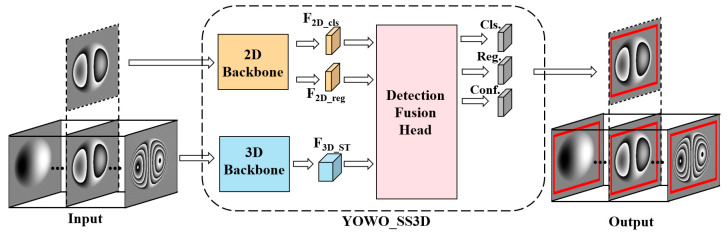
YOWO_SS3D network architecture.

**Figure 3 nanomaterials-15-00523-f003:**
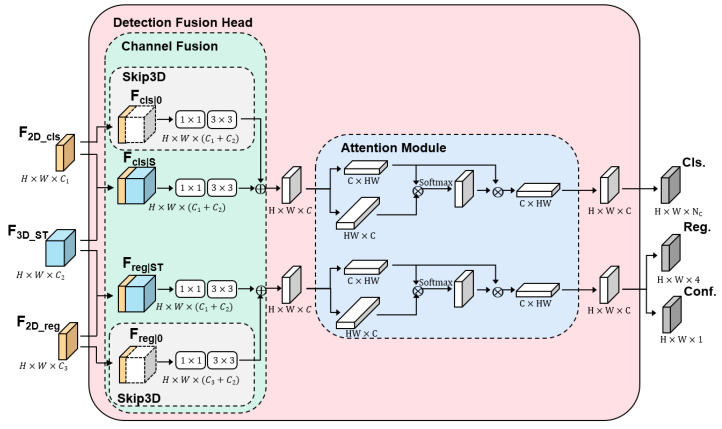
Detection fusion head with Skip3D.

**Figure 4 nanomaterials-15-00523-f004:**
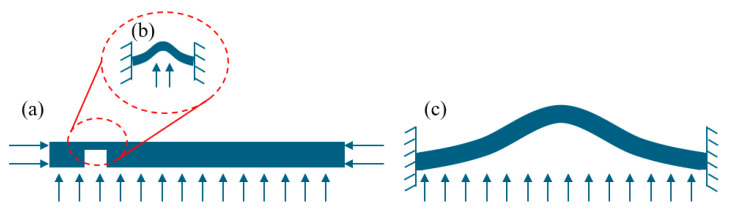
Plate theory approximation under pressure loading of the tested specimen. (**a**) Stress distribution under pressure loading of the tested specimen. (**b**) Plate theory approximation for the defect region. (**c**) Plate theory approximation for the entire tested specimen.

**Figure 5 nanomaterials-15-00523-f005:**
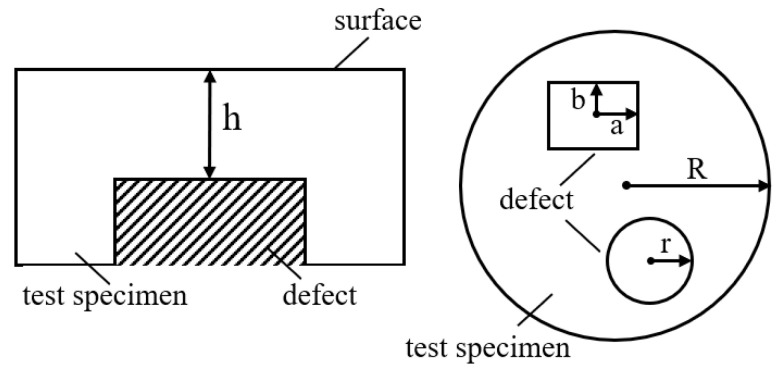
Definition of defect parameters in the tested specimen.

**Figure 6 nanomaterials-15-00523-f006:**
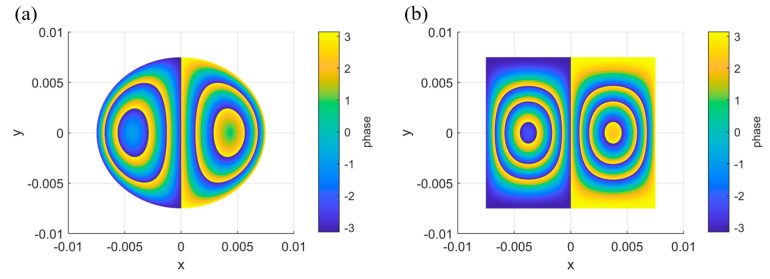
Typical phase map simulation results. (**a**) Simulation of abnormal fringes for the circular defect. (**b**) Simulation of abnormal fringes for the rectangular defect.

**Figure 7 nanomaterials-15-00523-f007:**
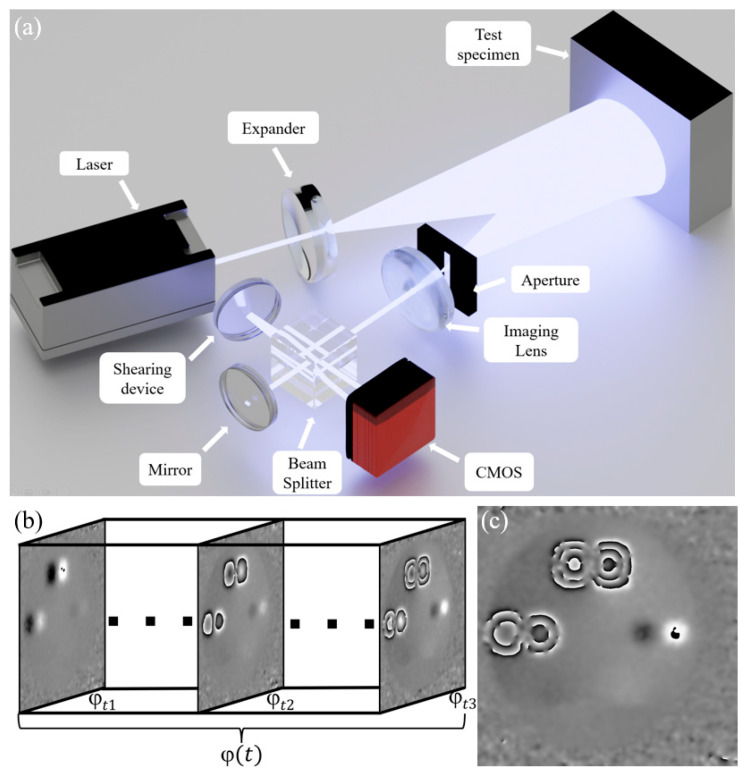
Experimental data acquisition. (**a**) Shearography system for dynamic detection. (**b**) Sequence of phase map experimental data. (**c**) Phase map experimental data.

**Figure 8 nanomaterials-15-00523-f008:**
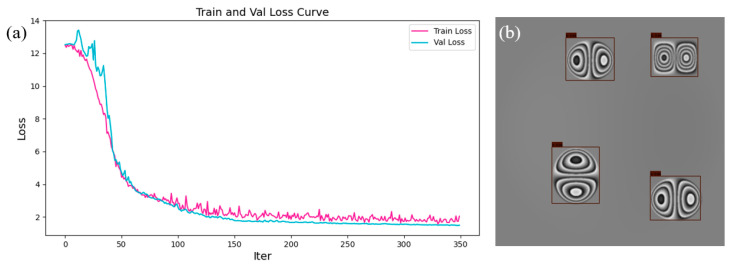
Training results of YOWO_SS3D on the simulation dataset. (**a**) Loss function curve during the training process. (**b**) Defect detection results on the simulated data test set.

**Figure 9 nanomaterials-15-00523-f009:**
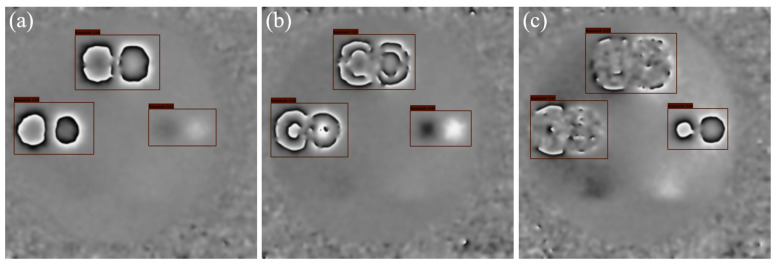
Typical defect detection results of YOWO_SS3D on experimental data under different external forces. (**a**) Small external force. (**b**) Moderate external force. (**c**) Large external force.

**Figure 10 nanomaterials-15-00523-f010:**
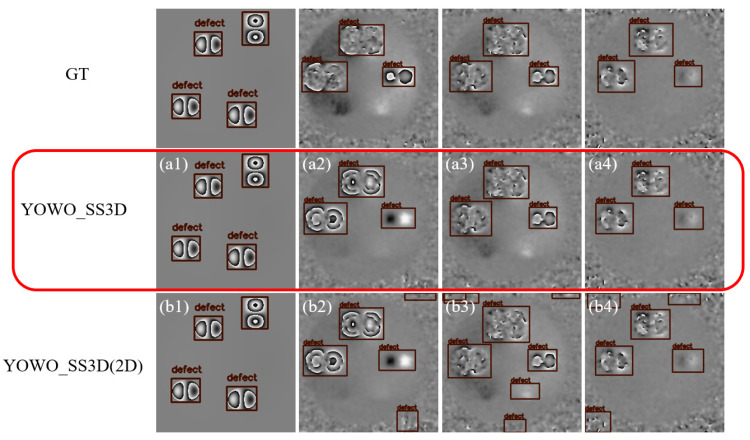
Defect detection results of different networks. (**a1**–**a4**) YOWO_SS3D network proposed in this study. (**b1**–**b4**) YOWO_SS3D(2D) network without 3D backbone.

**Table 1 nanomaterials-15-00523-t001:** Training parameter settings.

Epoch	Batch Size	Optimizer	Learning Rate	Loss Function
1	8	Adam	Linear warmup (0, 1 × 10^−4^)	$L\left(\left\{a_{x,y}\right\},\left\{b_{x,y}\right\},\left\{c_{x,y}\right\}\right)$
2–6			MultiStepLR (1 × 10^−4^, 6 × 10^−6^)	
7			6 × 10^−6^	

**Table 2 nanomaterials-15-00523-t002:** Impact of simulation dataset video parameters on the accuracy of the YOWO_SS3D network.

Group.	Simulation Dataset		Experimental Dataset		ACC
	Interval (mmHg)	Range (mmHg)	Interval (mmHg)	Range (mmHg)	
A	10	0–100	0.8–2.6	0–100	95.44%
B	6.6	0–100			95.88%
C	5	0–100			96.99%
D	4	0–100			95.45%
E	2	0–100			96.43%
F	1.5	0–100			95.79%
G	1	0–100			94.48%
H	5	0–120			94.32%
I	5	0–110			95.14%
J	5	0–90			95.47%
K	5	0–80			95.96%

**Table 3 nanomaterials-15-00523-t003:** Impact of 3D backbone on the performance of YOWO_SS3D.

Group.	Model	3D Backbone	ACC of Sim-Data	ACC of Exp-Data
A	YOWO_SS3D	Yes	100%	96.99%
B	YOWO_SS3D(2D)	No	100%	53.48%

**Table 4 nanomaterials-15-00523-t004:** Ablation experiment for the Skip3D feature fusion module.

Group.	Model	Skip3D	ACC of Sim-Data	ACC of Exp-Data
A	YOWO_SS3D	Yes	100%	96.99%
B	YOWO_SS3D(Skip3D_test)	No	100%	94.77%

**Table 5 nanomaterials-15-00523-t005:** Defect detection performance of YOLOv4 and YOWO_SS3D with limited experimental data training.

Group.	Train-Val Dataset		Test Dataset	ACC	
	Sim-Data	Exp-Data	Exp-Data	YOLOv4	YOWO_SS3D
A	0	100 (8:2)	1715	40.38%	80.96%
B		250 (8:2)		69.19%	88.47%
C		500 (8:2)		87.31%	89.20%
D		1000 (8:2)		84.46%	88.85%
E		1500 (8:2)		83.34%	89.53%

**Table 6 nanomaterials-15-00523-t006:** Defect detection performance of YOLOv4 and YOWO_SS3D under simulation–experimental data hybrid training.

Group.	Train-Val Dataset		Test Dataset	ACC	
	Sim-Data	Exp-Data	Exp-Data	YOLOv4	YOWO_SS3D
A	4000 (8:2)	0	1715	65.37%	96.99%
B	4000	100 (8:2)		86.75%	95.75%
C		250 (8:2)		90.74%	94.41%
D		500 (8:2)		90.01%	96.13%
E		1000 (8:2)		88.67%	96.14%
F		1500 (8:2)		86.49%	94.27%

## Data Availability

Data underlying the results presented in this paper are not publicly available at this time but may be obtained from the authors upon reasonable request.

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
