# Peer review of "Shearography-Based Near-Surface Defect Detection in Composite Materials: A Spatiotemporal Object Detection Neural Network Trained Only with Simulated Data"

_nanomaterials, 2025, doi:10.3390/nano15070523_

Round 1
Reviewer 1 Report
Comments and Suggestions for Authors Unfortunately this work did not convince me. It is not very clear to read and illustrates too many less interesting aspects. The experimental part for example is poorly discussed. I recommend a complete revision.Author Response
We really want to thank you for your critical review comments.
We revised the manuscript per the reviewers' comments. The responses to the comments are as follows. The responses are in the order of first explanations and then revisions, if available. The exact revision places are based on the revised manuscript, and the red lines are newly added sentences.
Comment 1:
Unfortunately, this work did not convince me. It is not very clear to read and illustrates too many less interesting aspects. The experimental part for example is poorly discussed. I recommend a complete revision.
Response 1:
We sincerely apologize that our work did not meet your expectations, and we believe this is due to our lack of clarity in the presentation. We organized the paper around the core contributions and innovations as follows, and we considered that all the sections were necessary and of interests to potential readers.
- Proposing a high-accuracy defect detection method utilizing temporal features to further enhance the spatial features of the defects. This approach captures the temporal similarities between simulated and experimental phase-map sequences, enabling the neural network to learn experimental characteristics of the defects from simulations. As a result, even when trained solely on simulation data, the proposed method achieves higher detection accuracy than traditional method which focuses only on the spatial feature of the defects in a phase map. The fundamental principles of the proposed method are described in Section 2.1, while the accuracy verification results are presented in Sections 3.1 and 3.2.
- Designing the YOWO_SS3D defect detection network, consists of a 3D backbone network and a 2D backbone network, which extract the temporal features from the phase map sequence and the spatial features from each phase map frame. Additionally, we designed the Skip3D feature fusion module to integrate these two types of features. Compared to YOLOv4 and other object detection networks that only contain a 2D backbone to extract spatial features, YOWO_SS3D leverages its 2D+3D backbone architecture to extract spatial features and temporal features, allowing it to effectively distinguish between noise and defects. With this advanced network architecture, YOWO_SS3D demonstrates exceptional defect detection performance, as discussed in Sections 3.2, 3.3, and 3.4.1. Furthermore, even when trained solely on simulation data, YOWO_SS3D achieves high-accuracy defect detection on experimental data, as demonstrated in Section 3.4.2.
- Providing a simulation approach for generating phase map sequences, which is achieved by mathematically modeling the entire shearography process. The modeling results accurately reflect the phase maps of defect regions observed in real experiments, making them suitable for network training. The principles of data simulation are detailed in Section 2.3.
For better emphasizing the above-mentioned contributions and better responding to your suggestions, we have conducted a comprehensive revision and kindly request your further critique and guidance.
- For better discussion of the experimental part, we have revised the "Results and Discussion" section based on your suggestions. To analyze the advantages of YOWO_SS3D over YOLOv4, we have added relevant analyses in Sections 3.4.1 and 3.4.2, as shown in the supplementary explanations around line 535 in Section 3.4.1 and around line 561 in Section 3.4.2.
Revision:
Line 535:
" We believe that the superior performance of YOWO_SS3D in the experiments can be attributed to its advanced network architecture and unique learning mechanism. In contrast, YOLOv4 is a typical 2D network that lacks the ability to learn temporal features. When trained solely on simulation data, YOLOv4 is more susceptible to noise interference, leading to a high rate of missed and false detections. By effectively leveraging both spatial and temporal features even with limited data, YOWO_SS3D demonstrates significantly superior defect detection capabilities compared to YOLOv4."
Line 561:
"During simulation-experimental data hybrid training, its detection accuracy remained stable around 95%, consistently outperforming YOLOv4. This indicates that YOWO_SS3D has superior defect detection performance compared to traditional single frame defect detection networks."
For clearer outline of the contribution and innovation, we have revised the "Abstract" and "Introduction" section.
- We have modified the description of YOLOv4 data in the abstract to more clearly highlight the differences between the YOWO_SS3D network and YOLOv4, as shown around line 22 of the revised manuscript.
Revision:
"The experimental results show that, with only 4000 frames of simulated data for training, our network achieved a detection accuracy of 96.99% on experimental phase maps, which is considerably higher than the 65.37% accuracy achieved by training the YOLOv4 network with the same simulated data.”
- We have revised the last two sentences of the abstract to emphasize the advantages of our method in terms of dataset dependency and its practical convenience in real-world detection tasks, as shown around line 24 of the revised manuscript.
Revision:
"Using our technique, only pre-generated simulation data is required to train the network, enabling YOWO_SS3D to be directly deployed for practical defect detection tasks. This approach eliminates the substantial costs associated with collecting experimental training data and promotes the application of neural network technology in the shearography field."
- We have added a brief explanation in the final paragraph of the introduction regarding why temporal features are beneficial for defect detection, as shown around line 102 of the revised manuscript. Additionally, we have made modifications in the experimental section (Section 3.3.1, around line 477) to further elaborate on the advantages brought by temporal features.
Revision:
Line 102:
"To eliminate the reliance on experimental data during the neural network training, while ensuring detection accuracy and reducing the cost of creating shearography datasets, this paper proposes a defect detection method using a spatio-temporal object detection network. This method learns defect features from both the spatial and temporal dimensions, allowing high-performance defect detection with simulation data alone for training."
Line 477:
" The ablation experiment clearly demonstrates that when trained on simulated data, the YOWO_SS3D network with the 3D module maintains high accuracy even on experimental data, despite the significant differences between simulated and experimental data. This is because the distinction between randomly varying noise and systematically changing defects is highly pronounced in temporal features. As a result, YOWO_SS3D, which leverages temporal features, achieves robust detection performance. This experiment proving the substantial benefit of temporal features for networks trained solely on simulated data."
- We have revised the transition between prior work and our study in the introduction to better clarify the research gap and the necessity of this study, as presented around line 89 of the revised manuscript.
Revision:
"Overall, although many methods exist to simulate phase maps, current approaches still only consider the spatial features of single frame phase map. A major challenge for neural networks is the significant gap between simulated and experimental data. This discrepancy arises because simulation data do not account for the inherent characteristics of specific hardware, such as the difficulty of accurately modeling complex material properties and environmental noise. As a result, networks trained solely on simulation data often exhibit lower accuracy on experimental data, for example the network may struggle to distinguish between noise and actual defects in experimental data, leading to missed or false detections. This limitation necessitates a heavy reliance on experimental data for training. However, collecting experimental data that comprehensively covers various test specimen materials, shapes, and defect types is challenging. This difficulty has become a major constraint on the generalization ability of deep neural networks."
Once again, we sincerely appreciate the time you have taken to review our manuscript. Your feedback has been immensely beneficial and has greatly improved the quality of our work. Thank you for your thoughtful and constructive comments.
Reviewer 2 Report
Comments and Suggestions for Authors
The manuscript presents a well-structured study on spatio-temporal defect detection in shearography. The proposed YOWO_SS3D network is an interesting approach that demonstrates strong performance in eliminating the reliance on experimental datasets for training.
The title is clear and informative, accurately reflecting the study's focus, but it is quite long. A more concise alternative could be: "Spatio-Temporal Neural Network for Shearography-Based Defect Detection in Composites Using Simulated Data". If the authors prefer the original title, it is still acceptable.
Abstract
The Abstract is clearly written and effectively summarizes the research.
It states that YOWO_SS3D achieves 96.99% accuracy, but does not explicitly compare this to other models (e.g., YOLOv4).
The term "spatio-temporal object detection network" might be too technical for general readers. A simpler explanation of how it improves defect detection is needed.
Dataset cost reduction is mentioned but does lacks stating its practical impact.
2. Introduction
The Introduction provides relevant background.
Better explain why using temporal changes in shearograms improves defect detection compared to single-frame analysis.
Clarify the research gap- prior work is well summarized, but it does not explicitly state what is missing and why this study is necessary.
The transition from prior work to the core research problem should be clearer. Explicitly state why current methods are insufficient.
List key contributions to improve readability and highlight the novelty of the approach.
3. Materials and Methods
The methodology is well-detailed, but some clarifications are needed.
Equation 1 is introduced without enough explanation for readers unfamiliar with the method.
Specify the number of unique specimens, defect variations, and control experiments conducted.
The choice of circular and rectangular defects should be explained. Do these shapes adequately represent real-world defects?
Clarify why certain hyperparameters were chosen and mention the GPU/hardware setup used for training.
4. Results and Discussion
The results are comprehensive, but further explanations are necessary.
It is stated that YOWO_SS3D outperforms YOLOv4, but the reason for this improvement is not analyzed. Elaborate on why spatio-temporal features make such a difference.
The drop in accuracy when removing the 3D backbone and Skip3D module is significant, but it is not fully explained why temporal features are so crucial.
Address potential weaknesses of the approach, such as generalization to unseen defect types and sensitivity to experimental variations.
It is unclear whether multiple runs were conducted to verify accuracy consistency. Report mean accuracy ± standard deviation.
The results are promising, but it is not discussed whether this method has been tested in industrial applications.
5. Conclusion
The conclusion effectively summarizes key findings.
Paraphrase the claim that the method "eliminates the need for experimental data", as experimental data were still needed for testing.
Briefly mention any remaining challenges before real-world application.
Provide suggestions for future work (testing on more defect types, optimizing computational efficiency, validating the model under different conditions).
The references cited are appropriate and relevant, covering shearography, defect detection, and deep learning with good balance between classic works and recent studies. Ensure sufficient coverage of alternative deep learning models, beyond YOLO-based approaches and completeness of citations, especially for shearography simulation and dataset generation methods.
Comments on the Quality of English LanguageThe English language in the manuscript is generally clear and understandable, with well-structured sentences and appropriate scientific terminology. There are instances of awkward phrasing, overly complex sentence structures, and minor grammatical errors. Some technical terms are introduced without sufficient explanation. Moderate editing is needed.
Author Response
We really want to thank you for your critical review comments.
We revised the manuscript per the reviewers' comments. The responses to the comments are as follows. The responses are in the order of first explanations and then revisions, if available. The exact revision places are based on the revised manuscript, and the red lines are newly added sentences.
Comment 1:
The title is clear and informative, accurately reflecting the study's focus, but it is quite long. A more concise alternative could be: "Spatio-Temporal Neural Network for Shearography-Based Defect Detection in Composites Using Simulated Data". If the authors prefer the original title, it is still acceptable.
Response 1:
Thank you very much for your valuable feedback. We appreciate your suggestion for a more concise title. However, we are concerned that the term "Spatio-Temporal Neural Network" might lead to potential confusion with other technologies such as spatio-temporal trajectory prediction, video classification, traffic flow prediction, and medical image time-series analysis. Therefore, we have decided to retain the original title. Once again, we sincerely thank you for your thoughtful suggestion.
Comment 2:
(Abstract) It states that YOWO_SS3D achieves 96.99% accuracy, but does not explicitly compare this to other models (e.g., YOLOv4).
Response 2:
Thank you very much for your valuable comments. We have modified the description of YOLOv4 data in the abstract to more clearly highlight the differences between the YOWO_SS3D network and YOLOv4, as shown around line 21 of the revised manuscript.
Revision 2:
"The experimental results show that, with only 4000 frames of simulated data for training, our network achieved a detection accuracy of 96.99% on experimental phase maps, which is considerably higher than the 65.37% accuracy achieved by training the YOLOv4 network with the same simulated data.”
Comment 3:
(Abstract) The term "spatio-temporal object detection network" might be too technical for general readers. A simpler explanation of how it improves defect detection is needed.
Response 3:
Thank you very much for your valuable comments. We have provided a more detailed explanation of the network input and the learning objectives of the "spatio-temporal object detection network," as presented around line 16 of the revised manuscript.
Revision 3:
"To address this issue, this paper utilizes phase map sequences measured by shearography as the medium for defect detection and designs a YOWO_SS3D spatio-temporal object detection network. The network simultaneously learns both the spatial distribution features and temporal variation patterns of simulated phase map sequences, achieving high-accuracy detection of defects."
Comment 4:
(Abstract) Dataset cost reduction is mentioned but does lacks stating its practical impact.
Response 4:
Thank you very much for your valuable comments. We have revised the last two sentences of the abstract to emphasize the advantages of our method in terms of dataset dependency and its practical convenience in real-world detection tasks, as shown around line 24 of the revised manuscript.
Revision 4:
"Using our technique, only pre-generated simulation data is required to train the network, enabling YOWO_SS3D to be directly deployed for practical defect detection tasks. This approach eliminates the substantial costs associated with collecting experimental training data and promotes the application of neural network technology in the shearography field."
Comment 5:
(Introduction) Better explain why using temporal changes in phase maps improves defect detection compared to single-frame analysis.
Response 5:
Thank you very much for your insightful comments. We have added a brief explanation in the final paragraph of the introduction regarding why temporal features are beneficial for defect detection, as shown around line 102 of the revised manuscript. Additionally, we have made modifications in the experimental section (Section 3.3.1, around line 477) to further elaborate on the advantages brought by temporal features.
Revision 5:
Line 102:
"To eliminate the reliance on experimental data during the neural network training, while ensuring detection accuracy and reducing the cost of creating shearography datasets, this paper proposes a defect detection method using a spatio-temporal object detection network. This method learns defect features from both the spatial and temporal dimensions, effectively eliminating the interference caused by random noise, allowing high-performance defect detection with simulation data alone for training."
Line 477:
" The ablation experiment clearly demonstrates that when trained on simulated data, the YOWO_SS3D network with the 3D module maintains high accuracy even on experimental data, despite the significant differences between simulated and experimental data. This is because the distinction between randomly varying noise and systematically changing defects is highly pronounced in temporal features. As a result, YOWO_SS3D, which leverages temporal features, achieves robust detection performance. This experiment proves the substantial benefit of temporal features for networks trained solely on simulated data."
Comment 6:
(Introduction) Clarify the research gap- prior work is well summarized, but it does not explicitly state what is missing and why this study is necessary.
The transition from prior work to the core research problem should be clearer. Explicitly state why current methods are insufficient.
Response 6:
Your feedback is highly valuable. We have revised the transition between prior work and our study in the introduction to better clarify the research gap and the necessity of this study, as presented around line 89 of the revised manuscript.
Revision 6:
"Overall, although many methods exist to simulate phase maps, current approaches still only consider the spatial features of single frame phase map. A major challenge for neural networks is the significant gap between simulated and experimental data. This discrepancy arises because simulation data do not account for the inherent characteristics of specific hardware, such as the difficulty of accurately modeling complex material properties and environmental noise. As a result, networks trained solely on simulation data often exhibit lower accuracy on experimental data, for example the network may struggle to distinguish between noise and actual defects in experimental data, leading to missed or false detections. This limitation necessitates a heavy reliance on experimental data for training. However, collecting experimental data that comprehensively covers various test specimen materials, shapes, and defect types is challenging. This difficulty has become a major constraint on the generalization ability of deep neural networks."
Comment 7:
(Introduction) List key contributions to improve readability and highlight the novelty of the approach.
Response 7:
Following your suggestion, we have included a summary of the key contributions of this study in the final paragraph of the introduction, as shown around line 128 of the revised manuscript.
Revision 7:
" Core contributions of this paper are summarized as follows
- Proposing a high-accuracy defect detection method utilizing temporal features to further enhance the spatial features of the defects. This approach captures the temporal similarities between simulated and experimental phase-map sequences, enabling the neural network to learn experimental characteristics of the defects from simulations. As a result, even when trained solely on simulation data, the proposed method achieves higher detection accuracy than traditional method which focuses only on the spatial feature of the defects in a phase map.
- Designing the YOWO_SS3D defect detection network, consists of a 3D backbone network and a 2D backbone network, which extract the temporal features from the phase map sequence and the spatial features from each phase map frame. Additionally, we designed the Skip3D feature fusion module to integrate these two types of features. Com-pared to YOLOv4 [9] and other object detection networks that only contain a 2D backbone to extract spatial features, YOWO_SS3D leverages its 2D+3D backbone architecture to extract spatial features and temporal features, allowing it to effectively distinguish between noise and defects. With this advanced network architecture, YOWO_SS3D demonstrates exceptional defect detection performance. Furthermore, even when trained solely on simulation data, YOWO_SS3D achieves high-accuracy defect detection on experimental data.
- Providing a simulation approach for generating phase map sequences, which is achieved by mathematically modeling the entire shearography process. The modeling results accurately reflect the phase maps of defect regions observed in real experiments, making them suitable for network training.
[9]. Weixian L., Dandan W., Sijin W., Simulation Dataset Preparation and Hybrid T raining for Deep Learning in Defect Detection Using Digital Shearography, Applied Sciences, 2022, 12, 6931."
Comment 8:
(Materials and Methods) Equation 1 is introduced without enough explanation for readers unfamiliar with the method.
Response 8:
Thank you very much for your valuable comments. Equation 1 is the fundamental formula for defect detection using shearography. We have provided a more detailed explanation of the variables in Equation 1 and their roles in defect detection, as described around line 153 of the revised manuscript.
Revision 8:
"Shearography measures the out-of-plane displacement gradient of the tested specimen by applying external force, with its phase distribution referred to as a phase map. Assuming that at time t, light with a wavelength λ is incident on a region experiencing an out-of-plane displacement wt, it undergoes diffuse reflection and forms lateral shearing interference within the optical system. The resulting phase map φt is determined by the phase distribution caused by the out-of-plane displacement and the phase shift introduced by lateral shearing. Thus, the phase difference is proportional to both the partial derivative of the out-of-plane displacement δwt/δ|d| and the shearing amount |d| , with 4π/λ as the conversion factor between displacement and the reflected phase, as shown in Equation (1)"
Comment 9:
(Materials and Methods) Specify the number of unique specimens, defect variations, and control experiments conducted.
Response 9:
We have just one test specimen. Moreover, as described in Section 2.2.2, we performed data augmentation by adjusting optical parameters, ultimately generating 3216 datasets. We have introduced control experiments at the beginning of the "Results and Discussion" section to guide readers, as shown around line 330 of the revised manuscript.
Revision 9:
"The test specimen, which includes square and circular defects representing common defect in composite materials such as void and delamination, were subjected to uniform air pressure loading using a pneumatic pump."
Comment 10:
(Materials and Methods) The choice of circular and rectangular defects should be explained. Do these shapes adequately represent real-world defects?
Response 10:
Thank you very much for your valuable comments.
We have provided a more detailed description of the standard specimens and explained how the circular and rectangular defects in these specimens represent real-world defects, thereby demonstrating the reliability of the standard specimens, as discussed around line 330 of the revised manuscript.
Revision 10:
"The test specimen, which includes square and circular defects representing common defect in composite materials such as void and delamination, were subjected to uniform air pressure loading using a pneumatic pump."
Comment 11:
(Materials and Methods) Clarify why certain hyperparameters were chosen and mention the GPU/hardware setup used for training.
Response 11:
Thank you very much for your valuable comments. We have revised the Materials and Methods section based on your suggestions. The hyperparameters provided in this paper are empirical values obtained through extensive testing. We fine-tuned the hyperparameters based on commonly used settings in object detection neural networks, conducting approximately 3–5 comparative tests on parameters such as batch size and learning rate. The results showed that within a reasonable range, hyperparameter variations had minimal impact on accuracy. Since hyperparameter selection is not the primary focus of this study, we did not elaborate on it in detail in the original text. Ultimately, we balanced training time and network accuracy to determine the final hyperparameters described in the manuscript. However, we have included a description of our hardware setup at the beginning of the "Results and Discussion" section to facilitate reproducibility, as shown around line 378 of the revised manuscript.
Revision 11:
"Additionally, we designed two comparison experiments to demonstrate the advantages of YOWO_SS3D over traditional methods, as shown in Sec. 3.4. All training experiments were conducted on a GPU device, specifically the RTX 4090D with 24GB of VRAM."
Comment 12:
(Results and Discussion) It is stated that YOWO_SS3D outperforms YOLOv4, but the reason for this improvement is not analyzed. Elaborate on why spatio-temporal features make such a difference.
Response 12:
Thank you very much for your valuable comments. We have revised the "Results and Discussion" section based on your suggestions. To analyze the advantages of YOWO_SS3D over YOLOv4, we have added relevant analyses in Sections 3.4.1 and 3.4.2, as shown in the supplementary explanations around line 535 in Section 3.4.1 and around line 561 in Section 3.4.2.
Revision 12:
Line 535:
"We believe that the superior performance of YOWO_SS3D in the experiments can be attributed to its advanced network architecture and unique learning mechanism. In contrast, YOLOv4 is a typical 2D network that lacks the ability to learn temporal features. When trained solely on simulation data, YOLOv4 is more susceptible to noise interference, leading to a high rate of missed and false detections. By effectively leveraging both spatial and temporal features even with limited data, YOWO_SS3D demonstrates significantly superior defect detection capabilities compared to YOLOv4."
Line 561:
"During simulation-experimental data hybrid training, its detection accuracy remained stable around 95%, consistently outperforming YOLOv4. This indicates that YOWO_SS3D has superior defect detection performance compared to traditional single frame defect detection networks. "
Comment 13:
(Results and Discussion) The drop in accuracy when removing the 3D backbone and Skip3D module is significant, but it is not fully explained why temporal features are so crucial.
Response 13:
Thank you very much for your valuable comments. We have revised the "Results and Discussion" section based on your suggestions. We previously did not clearly explain the relationship between the 3D backbone and temporal features. In fact, temporal features are extracted by the 3D backbone, so the experiments in Section 3.3.1 fully demonstrate the advantages brought by temporal features. We have added an explanation in Section 3.3.1, as described around line 443 of the revised manuscript.
We have also analyzed the differences in experimental results between YOWO_SS3D and YOWO_SS3D(2D) around line 475 of the revised manuscript.
Revision 13:
Line 443:
"To verify the contribution of the 3D backbone network module in the YOWO_SS3D network, we set the 3D weights in the feature fusion stage to 0, while keeping the other network structures unchanged, resulting in the YOWO_SS3D(2D) model. This modification effectively decouples spatio-temporal interactions, forcing the model to rely solely on spatial features. "
Line 475:
" This results in a significant number of false positives in the YOWO_SS3D(2D) network, such as the detection results at the edges of the images in Figs. 10 (b2)–(b4). The ablation experiment clearly demonstrates that when trained on simulated data, the YOWO_SS3D network with the 3D module maintains high accuracy even on experimental data, despite the significant noise differences between simulated and experimental data. This is because the distinction between randomly varying noise and systematically changing defects is highly pronounced in temporal features. As a result, YOWO_SS3D, which leverages temporal features, achieves robust detection performance. This experiment proves the substantial benefit of temporal features for networks trained solely on simulated data."
Comment 14:
(Results and Discussion) Address potential weaknesses of the approach, such as generalization to unseen defect types and sensitivity to experimental variations.
Response 14:
Thank you for your suggestions. We have added a new Section 3.5 to discuss the potential for industrial applications (corresponding to Comments 16) and the potential weaknesses of this study, as shown around line 565 of the revised manuscript.
Revision 14:
"3.5 Discussion
Based on experimental analysis, the YOWO_SS3D network demonstrates excellent performance in multiple aspects after being trained on simulated data. We believe that the YOWO_SS3D network holds significant potential for practical engineering applications, provided that the following aspects are emphasized. First, the optical devices used in practical engineering should have the capability for multi-frame continuous measurement to provide phase map sequences for detection. For example, using the spatial carrier method [21] for dynamic phase map calculation is recommended. Additionally, phase shifting method [22] can be employed if the deformation of the test object is controllable and stable, and the process is performed as quickly as possible. Second, the types of defects that may occur in the actual test objects should be estimated in advance to ensure that corresponding defect categories are included in the simulated dataset. Finally, to ensure optimal detection performance, the video parameters of the simulated dataset should be carefully aligned with those of the actual measurement process, minimizing any discrepancies that might arise. While this work demonstrates the potential of YOWO_SS3D, it does not include evaluations on complex defects (such as crack defects) or robustness testing in real industrial scenarios with multi-material substrates and surface interference. These aspects can be explored in future research to strengthen the practical applicability of the proposed method.
21 G Pedrini, Y-L Zou, H J Tiziani, “Quantitative evaluation of digital shearing interferogram using the spatial carrier method,” Pure and Applied Optics 5, 313-321(1996)
22 Y. H. Huang et al. “NDT&E using shearography with impulsive thermal stressing and clustering phase extraction,” Opt. Laser Eng. 47(7), 774–781 (2009). "
Comment 15:
(Results and Discussion) It is unclear whether multiple runs were conducted to verify accuracy consistency. Report mean accuracy ± standard deviation.
Response 15:
We sincerely appreciate your question. In fact, we conducted multiple preliminary experiments to verify reproducibility. However, to ensure the repeatability of the experiments, we fixed the random seed during the training phase, as described around line 340 of the revised manuscript. The main difference between the YOWO_SS3D network and traditional network models lies in its unique temporal feature learning mechanism. Therefore, random factors such as the number of frames and frame rates in actual industrial inspection video data are more critical parameters affecting reproducibility. We place great emphasis on the reproducibility of network accuracy and have conducted experiments as shown in Table 2. The final accuracy of YOWO_SS3D on the experimental data is 95.65% ± 1.33, as shown around line 429 of the revised manuscript.
Revision 15:
"Nevertheless, the accuracy remains 95.65%±1.33%, indicating that the YOWO_SS3D network, trained solely on simulation data, not only ensures high performance but also possesses the advantage of being less sensitive to the video parameters of the training dataset, which facilitates the creation of simulation datasets for industrial applications."
Comment 16:
(Results and Discussion) The results are promising, but it is not discussed whether this method has been tested in industrial applications.
Response 16:
Thank you for your suggestions. This method has not been tested in industrial applications, we have added a new Section 3.5 to discuss the potential for industrial applications and the potential weaknesses of this study (corresponding to Comment 14), as shown around line 565 of the revised manuscript.
Revision 16:
"3.5 Discussion
Based on experimental analysis, the YOWO_SS3D network demonstrates excellent performance in multiple aspects after being trained on simulated data. We believe that the YOWO_SS3D network holds significant potential for practical engineering applications, provided that the following aspects are emphasized. First, the optical devices used in practical engineering should have the capability for multi-frame continuous measurement to provide phase map sequences for detection. For example, using the spatial carrier method [21] for dynamic phase map calculation is recommended. Additionally, phase shifting method [22] can be employed if the deformation of the test object is controllable and stable, and the process is performed as quickly as possible. Second, the types of defects that may occur in the actual test objects should be estimated in advance to ensure that corresponding defect categories are included in the simulated dataset. Finally, to ensure optimal detection performance, the video parameters of the simulated dataset should be carefully aligned with those of the actual measurement process, minimizing any discrepancies that might arise. While this work demonstrates the potential of YOWO_SS3D, it does not include evaluations on complex defects (such as crack defects) or robustness testing in real industrial scenarios with multi-material substrates and surface interference. These aspects can be explored in future research to strengthen the practical applicability of the proposed method.
21 G Pedrini, Y-L Zou, H J Tiziani, “Quantitative evaluation of digital shearing interferogram using the spatial carrier method,” Pure and Applied Optics 5, 313-321(1996)
22 Y. H. Huang et al. “NDT&E using shearography with impulsive thermal stressing and clustering phase extraction,” Opt. Laser Eng. 47(7), 774–781 (2009). "
Comment 17:
(Conclusion) Paraphrase the claim that the method "eliminates the need for experimental data", as experimental data were still needed for testing.
Response 17:
Thank you very much for your valuable comments.
To avoid ambiguity, we have rephrased the claim that the method "eliminates the need for experimental data," as shown around line 587 of the revised manuscript.
Revision 17:
" To address the issue of limited availability of shearography datasets, this paper de-signs the YOWO_SS3D spatio-temporal object detection network, which combines the temporal features of phase map sequences with spatial features to perform near-surface defect detection. This approach achieves high-performance deep learning defect detection using only simulation data for training."
Comment 18:
(Conclusion) Briefly mention any remaining challenges before real-world application.
Response 18:
Thank you very much for your valuable comments. We believe that generalization testing may pose new challenges for YOWO_SS3D, so we have added a paragraph on remaining challenges and suggestions for future work (corresponding to Comment 19) at the end of the paper, as shown around line 606 of the revised manuscript.
Revision 18:
"To extend the practical impact of this work, two critical directions should be prioritized in future: 1) The test specimen model in this study is a simplified version, with defect types limited to only circular and square shapes. Future work should evaluate the model’s performance in handling defects with complex stress distributions (e.g., crack-type defects) 2) Validating detection robustness in real industrial environments involving multi-material substrates and surface interference. "
Comment 19:
(Conclusion) Provide suggestions for future work (testing on more defect types, optimizing computational efficiency, validating the model under different conditions).
Response 19:
Thank you very much for your valuable comments. We believe that testing more complex defects and analyzing the robustness in real-world detection environments are potential directions for future work. We have added a paragraph on remaining challenges (corresponding to Comment 18) and suggestions for future work at the end of the paper, as shown around line 606 of the revised manuscript.
Revision 19:
"To extend the practical impact of this work, two critical directions should be prioritized in future: 1) The test specimen model in this study is a simplified version, with defect types limited to only circular and square shapes. Future work should evaluate the model‘s performance in handling defects with complex stress distributions (e.g., crack-type defects) 2) Validating detection robustness in real industrial environments involving multi-material substrates and surface interference. "
Comment 20:
(Conclusion) The references cited are appropriate and relevant, covering shearography, defect detection, and deep learning with good balance between classic works and recent studies. Ensure sufficient coverage of alternative deep learning models, beyond YOLO-based approaches and completeness of citations, especially for shearography simulation and dataset generation methods.
Response 20:
Thank you very much for your valuable comments. We have checked the sufficient coverage and citation of alternative deep learning models and shearography simulation and dataset generation methods. Explanations are as follows. In addition to YOLOv4, we have introduced several neural network categories for shearography defect detection in the introduction, such as Fast R-CNN (Citation 2 in the original text), YOLOv3 (Citation 3 in the original text), texture feature extraction networks (Citation 8 in the original text), and semantic segmentation networks (Citation 10 in the original text). The shearography simulation methods are also described in detail, as shown around line 73 of the revised manuscript.
Once again, we sincerely appreciate the time you have taken to review our manuscript. Your feedback has been immensely beneficial and has greatly improved the quality of our work. Thank you for your thoughtful and constructive comments.
Reviewer 3 Report
Comments and Suggestions for Authors
The article is of great interest and relevance, and the subject matter is thoroughly contemporary. In the final version, I propose to analyse and incorporate the following comments, which should enhance its quality.
Firstly, it would be beneficial to summarise the most significant advantages of the method described in the article and its novelty at the conclusion of the Introduction section.
Additionally, the scope of application of the Shearography-Based Near-Surface Defect Detection method should be further defined.
Furthermore, the potential for practical and engineering applications of the method described should be explored. Is it possible to present a measurement procedure with key and possibly universal parameters in the specified range?
Finally, the most significant conclusions should be summarised in the form of a bulleted list.
Author Response
We really want to thank you for your critical review comments.
We revised the manuscript per the reviewers' comments. The responses to the comments are as follows. The responses are in the order of first explanations and then revisions, if available. The exact revision places are based on the revised manuscript, and the red lines are newly added sentences.
Comments 1:
Firstly, it would be beneficial to summarise the most significant advantages of the method described in the article and its novelty at the conclusion of the Introduction section.
Response 1:
Thank you very much for your valuable suggestion. Following your advice, we have added a summary of the core contributions, and the novelty of the method described in the article at the end of the Introduction section, as shown around line 128 of the revised manuscript.
Revision 1:
" Core contributions of this paper are summarized as follows
- Proposing a high-accuracy defect detection method utilizing temporal features to further enhance the spatial features of the defects. This approach captures the temporal similarities between simulated and experimental phase-map sequences, enabling the neural network to learn experimental characteristics of the defects from simulations. As a result, even when trained solely on simulation data, the proposed method achieves higher detection accuracy than traditional method which focuses only on the spatial feature of the defects in a phase map.
- Designing the YOWO_SS3D defect detection network, consists of a 3D backbone network and a 2D backbone network, which extract the temporal features from the phase map sequence and the spatial features from each phase map frame. Additionally, we designed the Skip3D feature fusion module to integrate these two types of features. Com-pared to YOLOv4 [9] and other object detection networks that only contain a 2D backbone to extract spatial features, YOWO_SS3D leverages its 2D+3D backbone architecture to extract spatial features and temporal features, allowing it to effectively distinguish between noise and defects. With this advanced network architecture, YOWO_SS3D demonstrates exceptional defect detection performance. Furthermore, even when trained solely on simulation data, YOWO_SS3D achieves high-accuracy defect detection on experimental data.
- Providing a simulation approach for generating phase map sequences, which is achieved by mathematically modeling the entire shearography process. The modeling results accurately reflect the phase maps of defect regions observed in real experiments, making them suitable for network training.
[9]. Weixian L., Dandan W., Sijin W., Simulation Dataset Preparation and Hybrid T raining for Deep Learning in Defect Detection Using Digital Shearography, Applied Sciences, 2022, 12, 6931."
Comments 2:
Additionally, the scope of application of the Shearography-Based Near-Surface Defect Detection method should be further defined.
Response 2:
Thank you very much for your insightful questions. Specifically, Shearography is highly effective for detecting near-surface defects in composite materials (CFRP, GFRP), metals (aluminum, titanium, stainless steel), ceramics, and polymer-based coatings. The technique is particularly sensitive to defects such as delaminations, debonding, cracks, voids, and impact damage, with a typical detection depth ranging from 0.1 mm to 3 mm, depending on material properties and loading conditions. Shearography has been widely applied in aerospace (aircraft fuselage, turbine blades), automotive (composite body panels), wind energy (turbine blades), and marine industries (hull structures), as well as in semiconductor packaging for microcrack detection.
In the Introduction, we have added specific shearography application scenarios to provide clearer context, as shown around line 33 of the revised manuscript. In Section 3.2, we explored the scope of application of the YOWO_SS3D network for near-surface defect detection. Specifically, we analyzed the impact of the number of video frames and frame rates on detection accuracy. The detailed experimental results and conclusions are presented in Section 3.2, as shown around line 400 of the revised manuscript. Additionally, we have added a new Section 3.5 to discuss the potential for industrial applications and the limitations of this study, as shown around line 565 of the revised manuscript.
Revision 2:
Line 33:
" In recent years, composite materials have been widely used in aerospace, automotive, wind energy, and marine industries due to their superior mechanical properties. However, near-surface defects such as delaminations, debonding, and cracks often occur during manufacturing and service, compromising structural integrity [1]. Among numerous near-surface defect detection methods, shearography has emerged as an outstanding optical inspection technique for composite materials, owing to its advantages of non-destructive testing and strong anti-interference capabilities [2,3,4]."
Line 565:
"3.5 Discussion
Based on experimental analysis, the YOWO_SS3D network demonstrates excellent performance in multiple aspects after being trained on simulated data. We believe that the YOWO_SS3D network holds significant potential for practical engineering applications, provided that the following aspects are emphasized. First, the optical devices used in practical engineering should have the capability for multi-frame continuous measurement to provide phase map sequences for detection. For example, using the spatial carrier method [21] for dynamic phase map calculation is recommended. Additionally, phase shifting method [22] can be employed if the deformation of the test object is controllable and stable, and the process is performed as quickly as possible. Second, the types of defects that may occur in the actual test objects should be estimated in advance to ensure that corresponding defect categories are included in the simulated dataset. Finally, to ensure optimal detection performance, the video parameters of the simulated dataset should be carefully aligned with those of the actual measurement process, minimizing any discrepancies that might arise. While this work demonstrates the potential of YOWO_SS3D, it does not include evaluations on complex defects (such as crack defects) or robustness testing in real industrial scenarios with multi-material substrates and surface interference. These aspects can be explored in future research to strengthen the practical applicability of the proposed method."
21 G Pedrini, Y-L Zou, H J Tiziani, “Quantitative evaluation of digital shearing interferogram using the spatial carrier method,” Pure and Applied Optics 5, 313-321(1996)
22 Y. H. Huang et al. “NDT&E using shearography with impulsive thermal stressing and clustering phase extraction,” Opt. Laser Eng. 47(7), 774–781 (2009). "
Comments 3:
Furthermore, the potential for practical and engineering applications of the method described should be explored.
Response 3:
Thank you very much for your insightful questions. we have added a new Section 3.5 to discuss the potential for industrial applications and the limitations of this study, as shown around line 565 of the revised manuscript. What’s more, we have added a section on prospects at the end of the article, as shown around line 606 of the revised manuscript.
Revision 3:
Line 565:
"3.5 Discussion
Based on experimental analysis, the YOWO_SS3D network demonstrates excellent performance in multiple aspects after being trained on simulated data. We believe that the YOWO_SS3D network holds significant potential for practical engineering applications, provided that the following aspects are emphasized. First, the optical devices used in practical engineering should have the capability for multi-frame continuous measurement to provide phase map sequences for detection. For example, using the spatial carrier method [21] for dynamic phase map calculation is recommended. Additionally, phase shifting method [22] can be employed if the deformation of the test object is controllable and stable, and the process is performed as quickly as possible. Second, the types of defects that may occur in the actual test objects should be estimated in advance to ensure that corresponding defect categories are included in the simulated dataset. Finally, to ensure optimal detection performance, the video parameters of the simulated dataset should be carefully aligned with those of the actual measurement process, minimizing any discrepancies that might arise. While this work demonstrates the potential of YOWO_SS3D, it does not include evaluations on complex defects (such as crack defects) or robustness testing in real industrial scenarios with multi-material substrates and surface interference. These aspects can be explored in future research to strengthen the practical applicability of the proposed method."
21 G Pedrini, Y-L Zou, H J Tiziani, “Quantitative evaluation of digital shearing interferogram using the spatial carrier method,” Pure and Applied Optics 5, 313-321(1996)
22 Y. H. Huang et al. “NDT&E using shearography with impulsive thermal stressing and clustering phase extraction,” Opt. Laser Eng. 47(7), 774–781 (2009). "
Line 606:
"To extend the practical impact of this work, two critical directions should be prioritized in future: 1) The test specimen model in this study is a simplified version, with defect types limited to only circular and square shapes. Future work should evaluate the model’s performance in handling defects with complex stress distributions (e.g., crack-type defects) 2) Validating detection robustness in real industrial environments involving multi-material substrates and surface interference. "
Comments 4:
Is it possible to present a measurement procedure with key and possibly universal parameters in the specified range?
Response 4:
Thank you very much for your insightful questions. Due to the limitations of our current dataset and proposed method, we are unable to provide universal parameters for a wide range of scenarios.
We have not conducted extensive generalization testing or industrial application evaluations, and we believe these are important directions for future research. We have included a related analysis in the newly added Section 3.5 (around line 565) and provided a future outlook at the end of the paper (around line 606).
Revision 4:
Line 565:
"3.5 Discussion
Based on experimental analysis, the YOWO_SS3D network demonstrates excellent performance in multiple aspects after being trained on simulated data. We believe that the YOWO_SS3D network holds significant potential for practical engineering applications, provided that the following aspects are emphasized. First, the optical devices used in practical engineering should have the capability for multi-frame continuous measurement to provide phase map sequences for detection. For example, using the spatial carrier method [21] for dynamic phase map calculation is recommended. Additionally, phase shifting method [22] can be employed if the deformation of the test object is controllable and stable, and the process is performed as quickly as possible. Second, the types of defects that may occur in the actual test objects should be estimated in advance to ensure that corresponding defect categories are included in the simulated dataset. Finally, to ensure optimal detection performance, the video parameters of the simulated dataset should be carefully aligned with those of the actual measurement process, minimizing any discrepancies that might arise. While this work demonstrates the potential of YOWO_SS3D, it does not include evaluations on complex defects (such as crack defects) or robustness testing in real industrial scenarios with multi-material substrates and surface interference. These aspects can be explored in future research to strengthen the practical applicability of the proposed method."
21 G Pedrini, Y-L Zou, H J Tiziani, “Quantitative evaluation of digital shearing interferogram using the spatial carrier method,” Pure and Applied Optics 5, 313-321(1996)
22 Y. H. Huang et al. “NDT&E using shearography with impulsive thermal stressing and clustering phase extraction,” Opt. Laser Eng. 47(7), 774–781 (2009). "
Line 606:
"To extend the practical impact of this work, two critical directions should be prioritized in future: 1) The test specimen model in this study is a simplified version, with defect types limited to only circular and square shapes. Future work should evaluate the model’s performance in handling defects with complex stress distributions (e.g., crack-type defects) 2) Validating detection robustness in real industrial environments involving multi-material substrates and surface interference. "
Comments 5:
Finally, the most significant conclusions should be summarised in the form of a bulleted list.
Response 5:
Thank you very much for your suggestion. We have summarized the core contributions of this study in a bulleted list at the end of the Introduction section, as shown around line 128 of the revised manuscript. Furthermore, we have outlined future research directions in a bulleted list in the Conclusion section, as shown around line 606 of the revised manuscript. We hope that these lists will help readers better understand the key contributions and future prospects of our work.
Revision 5:
Line 128
" Core contributions of this paper are summarized as follows
- Proposing a high-accuracy defect detection method utilizing temporal features to further enhance the spatial features of the defects. This approach captures the temporal similarities between simulated and experimental phase-map sequences, enabling the neural network to learn experimental characteristics of the defects from simulations. As a result, even when trained solely on simulation data, the proposed method achieves higher detection accuracy than traditional method which focuses only on the spatial feature of the defects in a phase map.
- Designing the YOWO_SS3D defect detection network, consists of a 3D backbone network and a 2D backbone network, which extract the temporal features from the phase map sequence and the spatial features from each phase map frame. Additionally, we designed the Skip3D feature fusion module to integrate these two types of features. Com-pared to YOLOv4 [9] and other object detection networks that only contain a 2D backbone to extract spatial features, YOWO_SS3D leverages its 2D+3D backbone architecture to extract spatial features and temporal features, allowing it to effectively distinguish between noise and defects. With this advanced network architecture, YOWO_SS3D demonstrates exceptional defect detection performance. Furthermore, even when trained solely on simulation data, YOWO_SS3D achieves high-accuracy defect detection on experimental data.
- Providing a simulation approach for generating phase map sequences, which is achieved by mathematically modeling the entire shearography process. The modeling results accurately reflect the phase maps of defect regions observed in real experiments, making them suitable for network training.
[9]. Weixian L., Dandan W., Sijin W., Simulation Dataset Preparation and Hybrid T raining for Deep Learning in Defect Detection Using Digital Shearography, Applied Sciences, 2022, 12, 6931."
Line 606
"To extend the practical impact of this work, two critical directions should be prioritized in future: 1) The test specimen model in this study is a simplified version, with defect types limited to only circular and square shapes. Future work should evaluate the model’s performance in handling defects with complex stress distributions (e.g., crack-type defects) 2) Validating detection robustness in real industrial environments involving multi-material substrates and surface interference. "
Once again, we sincerely appreciate your constructive feedback, which has significantly improved the clarity and structure of our manuscript. Thank you for your thoughtful comments.
Round 2
Reviewer 2 Report
Comments and Suggestions for Authors
-